# Using national virtual maternal death reviews to improve the quality of care during Pregnancy, labour and birth, and Postpartum in Tanzania

Ahmad Mohamed Makuwani [1]*, Sunday Alfred Dominico[1,2]¶,
Secilia Kapalata Ngweshemi[3], Golden Mwakibo Masika[3], Rose Mpembeni[4],
Charles Ameh[5], Mzee Masumbuko Nassoro[1], Phineas Sospeter[1], Habib Ismail[1],
Tumaini Nagu[1], Matilda Ngarina[2], Edwin Lugazia[6], Pius Muzzazzi[7], Beatrice Mwilike[4,8],
Florina Serbanescu[9], for the Tanzania Virtual MPDSR Steering Committee¶

1 Ministry of Health, 2 Association of Obstetricians and Gynaecologists of Tanzania (AGOTA),
3 University of Dodoma, Dodoma, Dar Es Salaam, Tanzania, 4 Muhimbili University of Health and
Allied Sciences, Salaam, Tanzania, 5 Liverpool School of Tropical Medicine, Liverpool, United Kingdom,
6 Society of Anaesthesiologists of Tanzania (SATA), 7 Paediatric Association of Tanzania (PAT),
8 Tanzania Midwives Association (TAMA), 9 U.S. Centers for Disease Control and Prevention, Division of
Reproductive Health, Atlanta, Georgia, United States of America

¶ Membership of the Tanzania Virtual MPDSR Steering Committee is provided in the Acknowledgments.
* amakuwany@gmail.com

## Abstract

### Background

Tanzania has implemented the Maternal and Perinatal Death Surveillance and Response (MPDSR) system for more than a decade. However, multiple assessments have shown that the quality of maternal death reviews at facility and district levels has often been limited by inadequate specialist participation, weak root-cause analysis, and insufficient follow-through on action plans. These limitations have reduced the effectiveness of MPDSR in driving quality improvement and preventing avoidable maternal deaths. To address these gaps and strengthen accountability and clinical learning, the Ministry of Health introduced daily Virtual Maternal Death Reviews (VMDR) in 2021. This study describes the implementation and outcomes of VMDR from 2022 to 2023.

### Methods

A descriptive observational study was conducted using a mixed-methods approach. The analysis included maternal deaths that were notified through the national MPDSR system and subsequently selected for VMDR sessions between January 2022 and December 2023. Cases were eligible if they met the WHO definition of a maternal death and had sufficient clinical documentation from facility or district review committees to allow determination of the cause of death and contributing factors. Quantitative data were summarised using descriptive statistics, while qualitative

**Data availability statement:** "All relevant data are within the paper and its Supporting Information files.".

**Funding:** The author(s) received no specific funding for this work.

**Competing interests:** The authors have declared that no competing interests exist.

insights from VMDR discussions were synthesised to identify modifiable clinical and system-level factors and response actions.

## Results

A total of 369 VMDR sessions were conducted, reviewing 687 maternal deaths across all regions. The leading causes of death were obstetric haemorrhage (49%), hypertensive disorders of pregnancy (16%), and anaesthesia-related complications (10%). Common modifiable contributing factors included inadequate clinical competency (82%), suboptimal provider practices and attitudes (69%), weak leadership and accountability (42%), and gaps in surgical and anaesthesia care. VMDR participation facilitated improved oversight and prompted remedial actions such as redeployment of specialists, improvement of essential supplies, and initiation of structured mentorship and continuing medical education.

## Conclusion

The VMDR model strengthened the quality and consistency of maternal death reviews and improved accountability in responding to identified gaps. Sustaining VMDR requires continued leadership to maintain confidentiality in line with existing culture and norms.

---

## Introduction

Globally, an estimated 260,000 maternal deaths occurred in 2023 [1]. Despite accounting for only 10% of the world's population [2], Sub-Saharan Africa contributes 70% of these deaths (212,000 maternal deaths) [1]. In Tanzania, the maternal mortality ratio is estimated at 104 maternal deaths per 100,000 live births, based on the 2022 Tanzania Demographic and Health Survey and Malaria Indicator Survey. Although this represents a decline compared with previous estimates, it still translates to more than 2,000 maternal deaths annually, indicating a persistently high burden of preventable mortality [3]. Most maternal deaths in SSA occur during or within 24 hours of childbirth [4–6]. An increasing number of births in the region occur in health facilities [5]. Thus, improving the quality of delivery and post-delivery care in health facilities is a critical intervention to reduce maternal mortality [7–9] and achieve Sustainable Development Goal 3.1, which aims to reduce the global maternal mortality ratio to no more than 70 maternal deaths per 100,000 live births by 2030 [10].

Maternal death reviews (MDR) are key interventions that aim to identify and address gaps within the health care system that contributed to the death of a woman during pregnancy, delivery or the first 42 days after delivery [11]. Tanzania has been and continues to be among the top 10 countries with a high number of maternal deaths per year [12]. In 2006, Tanzania implemented maternal and perinatal death reviews for the first time, as recommended by the World Health Organisation (WHO) guidance *Beyond the Numbers, Reviewing Maternal Deaths, and Complications to Make Pregnancy Safer* [11]. The MDR process in Tanzania was scaled up into a

surveillance system following the publication of the WHO Maternal Death Surveillance and Response (MDSR) technical guidance [13,14]. This was captured in the first national guidelines issued in 2015 and revised in 2019 [15]. As with other countries in the region, Tanzania has extended the surveillance system to include perinatal deaths [15]. The MPDSR guidelines are intended to help eliminate preventable deaths in both facilities and communities by continuously documenting and analysing the magnitude, trends, and impact of maternal mortality. The guidelines prioritise a focus on strengthening accountability in responding to gaps in health systems, investigation of community deaths through verbal autopsies, and use of the WHO applications of ICD-10 to deaths during pregnancy, childbirth, and the puerperium (ICD-MM) and deaths during the perinatal period (ICD-PM) [11–13]. The guidelines provide ground rules for conducting death reviews, which include maintaining an atmosphere of "no shame, no blame, no name," respecting the confidentiality of the discussion and accepting criticisms from peers to improve clinical care [15].

Tanzania MPDSR guidelines are fully institutionalised at the community, health facility, district, regional, and national levels. For a meaningful critical MDR process, highly skilled clinicians organised in health facility committees must be present during the review. These professionals are generally available in hospitals (mainly regional, zonal and national), often in towns and cities. Several assessments of the national MPDSR guidelines implementation indicated delays and suboptimal quality in conducting reviews in lower-level health facilities [16,17]. Delays and suboptimal quality reviews were partly the result of a lack of commitment among health managers, time constraints, and limited training of reviewers [16–20]. Sub-optimal reviews led to incorrect assignment of causes of death, weak root-cause analysis (RCA) to identify gaps in the quality of care, and a failure to develop appropriate action plans that include specific, measurable, attainable, realistic, and time-bound (SMART) recommendations [21].

Experts recommend that MDSR be institutionalised within the Ministry of Health (MoH) to provide oversight, policy guidance, and support to ensure a functional and effective process. As a response to the COVID-19 pandemic, the MoH instituted limited physical travel for review meetings to mitigate the spread of the virus. This led to the adoption of virtual social networking applications such as Zoom, WhatsApp, Microsoft Teams, and Google Meet to facilitate technical interactions and support clinical decision-making for junior healthcare providers.

Despite the nationwide institutionalisation of MPDSR in Tanzania, challenges persisted in achieving the quality, depth and consistency required for effective maternal death reviews, especially in facilities with limited specialist involvement. To address these gaps, the Ministry of Health introduced the Virtual Maternal Death Review (VMDR) model in late 2021 as a health system innovation that strengthened the review process by using real-time digital platforms to bring together multidisciplinary expertise from across the country. VMDR enabled joint clinical analysis, facilitated collaboration between specialists regardless of location, improved efficiency through streamlined scheduling and digital record-sharing, promoted shared learning, established a robust platform for providing unbiased reviews and increased the accuracy of identifying modifiable factors. Since implementation in 2022–2023, the model has expanded access to specialist input, enhanced accountability among regional and district teams and accelerated the translation of review findings into targeted mentorship, supportive supervision and timely system-level actions. Through these improvements, VMDR represents an innovative advancement of the existing MPDSR process. This paper describes the implementation of VMDR in Tanzania, the patterns and modifiable factors identified, and the actions taken to improve the quality of care. As more countries seek to optimize their maternal mortality surveillance systems, learning from the experience of Tanzania may support an excellent opportunity to introduce the widespread usage of online review meetings in similar settings.

## Methods

### Design

A descriptive observational cross-sectional study was conducted using mixed methods. Quantitative data were used to summarise the distribution and characteristics of maternal deaths, while qualitative data were used to interpret the contributing factors during the review discussions.

### Settings and study population

The study covered maternal deaths reviewed during Virtual Maternal Death Review (VMDR) sessions conducted from 1 January 2022–31 December 2023 across all 26 regions of mainland Tanzania. All reviewed deaths met the WHO definition of a maternal death [ 22,23]. Cases were included when documentation from the facility and district reviews was sufficient to reconstruct the clinical course of care and identify contributing factors.

### Conceptual and organisational framework

VMDRs were conducted in alignment with MPDSR principles of confidentiality and no name, no blame, no shame [15,24]. The approach was informed by Bardwick's comfort zone concept, recognising that structured, professionally moderated review environments can promote accountability and learning without creating punitive pressure [25]. VMDR implementation was coordinated by the Ministry of Health in collaboration with the President's Office, Regional Administration and Local Government and professional associations. A national VMDR steering committee oversaw scheduling, standards of review, and rotation of session chairs. A national secretariat supported case preparation, documentation and follow-up.

### VMDR case review process

VMDR sessions were conducted via Zoom from Monday to Thursday each week (virtual perinatal death reviews were conducted on Friday). Each month, the Ministry of Health circulated a schedule specifying presenting regions, session chairs and supporting coordinators. The presenting region selected a maternal death that had already been reviewed at the facility and district levels and submitted the case summary for pre-assessment two to three days before the scheduled session.

Attendance typically ranged from 100 to over 300 participants, including facility clinicians, members of Council and Regional Health Management Teams, national programme managers and clinical experts. Attendance by senior national health leaders, including the Minister for Health, Deputy Minister, Permanent Secretary or Chief Medical Officer, occurred periodically and reinforced the seriousness of implementation follow-up. Attendance was mandatory for the Regional Medical Officer, District Medical Officer and the Medical Officer in-Charge of the facility where the death occurred, reflecting their supervisory and implementation responsibilities.

At the start of each session, the chair reminded participants of confidentiality and MPDSR conduct principles [15,24]. The presenting region first reported on follow-up of recommendations from prior VMDRs. The selected case was then presented using an anonymised narrative, with names replaced by initials or coded identifiers to protect privacy. The presentation followed a structured format from the original review: background, sequence of clinical events, investigations and treatments, decision points and referral actions.

The chair guided the discussion to reach consensus on the cause of death using the WHO ICD-MM classification [12] and facilitated a structured root-cause analysis to identify modifiable medical and system-level factors [26]. Identified gaps were reviewed in relation to the three delays framework with greater focus on delays in facility-based care and referral processes [27], given the high facility delivery rate in Tanzania [3]. Recommendations and action plans were developed with clear responsible actors and timelines and recorded for follow-up. Outcomes of VMDRs are not permitted to be used for reprimand, disciplinary action or litigation, maintaining the protected review environment.

### Primary data sources

The VMDR case documentation template, completed by the Regional Nursing Officer or Reproductive and Child Health Coordinator before and during the session, captured demographic information, obstetric history, the sequence of clinical events, investigations and treatments, final cause of death, modifiable factors and agreed recommendations. National MPDSR monthly notification reports provided surveillance data, including the distribution of maternal deaths by facility

level and region. Records from the VMDR Zoom session proceedings, including case presentations, discussion summaries and the chair's concluding remarks, documented the clinical reasoning informing case classification and decision-making. Where documentation was incomplete, clarification was sought from the involved facility before the review.

## Data management and analysis

Data from the VMDR documentation template and MPDSR notification forms were compiled in Microsoft Excel. Quantitative variables, including age, parity, place of death and cause of death, were summarised using descriptive statistics. Causes of death were categorised using WHO ICD-MM groupings [22]. Qualitative summaries and recorded discussion notes were analysed using deductive content analysis in Atlas.ti. Modifiable factors were categorised into eight domains: clinical skills; provider practices and attitudes; leadership and accountability; antenatal care; anaesthesia safety; surgical care; availability and use of blood and blood products; and availability of essential medicines and supplies (Annex 2). The gaps were tallied in an Excel sheet to establish how often they occurred across the reviewed deaths. Recommendations were synthesised by level of responsibility (facility, district, regional or national). Zoom licensing and data support were provided by the Ministry of Health with assistance from partners.

## Ethical considerations

VMDR activities form part of routine MPDSR quality improvement. All records were anonymised before analysis and archived at the Ministry of Health – Department of Reproductive, Maternal, and Child Health Services. These data do not bear the deceased's name and are kept confidential.

The Permission to publish these results was granted by the University of Dodoma (UDOM) Institutional Research Review Committee (IRREC) and approved by the National Health Research Ethics Sub-Committee (NatHREC) of the National Institute for Medical Research (NIMR) with reference number MA.84/261/67/16 after seeking clearance from the Ministry of Health.

## Results

A total of 369 VMDRs were conducted, and 687 maternal deaths that occurred from January 2022 to December 2023 were virtually reviewed. Reviewed cases were selected from maternal deaths that occurred in zonal referral hospitals (n = 151; 22%), regional referral hospitals (n = 159; 23%), district hospitals (n = 179; 26%), health centres (n = 136; 20%), dispensaries (n = 42; 6%), and communities (n = 20; 3%). Almost one-half of the 3,041 deaths notified to the MoH were from 7 regions (Dar es Salaam, Mwanza, Morogoro, Kigoma, Mtwara, Mbeya, and Tabora) (Table 1). Almost one-fourth (22.6%) of notified maternal deaths in 2022–2023 have been reviewed through VMDRs. The percentage of VMDR-reviewed deaths ranged from 11.3% in Dar es Salaam to 59.7% in the Pwani region. Because VMDR sessions follow a structured rotation in which each region is allocated scheduled opportunities to present cases regardless of the absolute number of maternal deaths, regions with fewer notified deaths were able to have a higher proportion of their cases reviewed virtually. This reflects the design of the review schedule rather than the preferential selection of regions.

The mean age of the 687 women who died of maternal causes and were VMDR-reviewed was 29.5 years, with a range of 15–50 years; 9% of women were less than 20 years old at the time of death.

Of the 687 maternal deaths VMDR-reviewed, 90% were due to direct obstetric causes, and 9% were due to indirect causes. The leading direct causes were obstetric haemorrhage (49%), hypertensive disorders of pregnancy (16%), anaesthesia-related complications (10%), puerperal sepsis (6%), obstetric embolism (4%), and abortion-related complications (3%). The leading indirect obstetric causes were anaemia (5%), malaria (2.6%), and preexisting cardiac conditions (1.5%). The cause of death was not established in 1% of cases due to inadequate information during the review (Fig 1).

The leading non-medical factors contributing to the maternal deaths reviewed through VMDRs were related to insufficient clinical skills (82%), healthcare providers' attitudes (69%), suboptimal leadership and accountability (42%),

**Table 1. Number of notified maternal deaths and proportion of virtually reviewed maternal deaths by region in 2022–2023, Tanzania Mainland.**

| S/N | Region | No. of MDs notified to MOH | No. of cases reviewed virtually | % MD of reviewed virtually |
|---|---|---|---|---|
| 1. | Dar es Salaam | 382 | 43 | 11.3% |
| 2. | Mwanza | 309 | 52 | 16.8% |
| 3. | Morogoro | 204 | 31 | 15.2% |
| 4. | Kigoma | 178 | 31 | 17.4% |
| 5. | Mtwara | 151 | 36 | 23.8% |
| 6. | Mbeya | 140 | 39 | 27.9% |
| 7. | Tabora | 132 | 22 | 16.7% |
| 8. | Dodoma | 123 | 20 | 16.3% |
| 9. | Geita | 112 | 22 | 19.6% |
| 10. | Singida | 111 | 25 | 22.5% |
| 11. | Arusha | 101 | 22 | 21.8% |
| 12. | Kagera | 98 | 24 | 24.5% |
| 13. | Mara | 95 | 16 | 16.8% |
| 14. | Shinyanga | 94 | 23 | 24.5% |
| 15. | Rukwa | 88 | 26 | 29.5% |
| 16. | Manyara | 83 | 25 | 30.1% |
| 17. | Kilimanjaro | 76 | 24 | 31.6% |
| 18. | Pwani | 72 | 43 | 59.7% |
| 19. | Songwe | 71 | 26 | 36.6% |
| 20. | Lindi | 70 | 21 | 30.0% |
| 21. | Katavi | 69 | 19 | 27.5% |
| 22. | Iringa | 67 | 29 | 43.3% |
| 23. | Simiyu | 62 | 22 | 35.5% |
| 24. | Tanga | 59 | 20 | 33.9% |
| 25. | Ruvuma | 49 | 15 | 30.6% |
| 26. | Njombe | 45 | 11 | 24.4% |
| | Overall | 3,041 | 687 | 22.6% |

inadequate surgical care (37%), inadequate antenatal care (34%), lack of blood (32%), poor anaesthesia practices (28%), client and community factors(27%) and lack of equipment and supplies (24%) (Fig 2). These causes often overlapped, each case having multiple contributing factors.

The VMDRs recommended specific actions (responses) to address avoidable factors related to each maternal death. Most agreed actions were related to 1) increased capacity of staff to manage obstetric complications that included improving surgical skills, improving consultations and mentorship (e.g., tertiary hospitals to improve multi-disciplinary consultations), and improving labour and postpartum monitoring; 2) adherence to the use of guidelines and standard operating procedures; and 3) increasing blood collection, processing, distribution and storage systems. This went hand in hand with the responsibilities of Regional Health Management Teams (RHMTS) and Council Health Management Teams (CHMTS) to ensure the functionality of the health system at primary healthcare levels, including the community.

In response to the VMDRs, RHMTs and CHMTs set up expert teams to build capacity based on identified gaps. Examples of the actions taken by the expert teams included safeguarding correct anaesthesia practices at one of the district hospitals, ensuring that specialist doctors in one of the zonal referral hospitals started a full-time night shift, connecting regional doctors to mentors in a zonal referral hospital to brush competencies; relocation of an anaesthesia machine to another health facility; and commencement of onsite mentorship to address key identified dysfunctions. The VMDRs

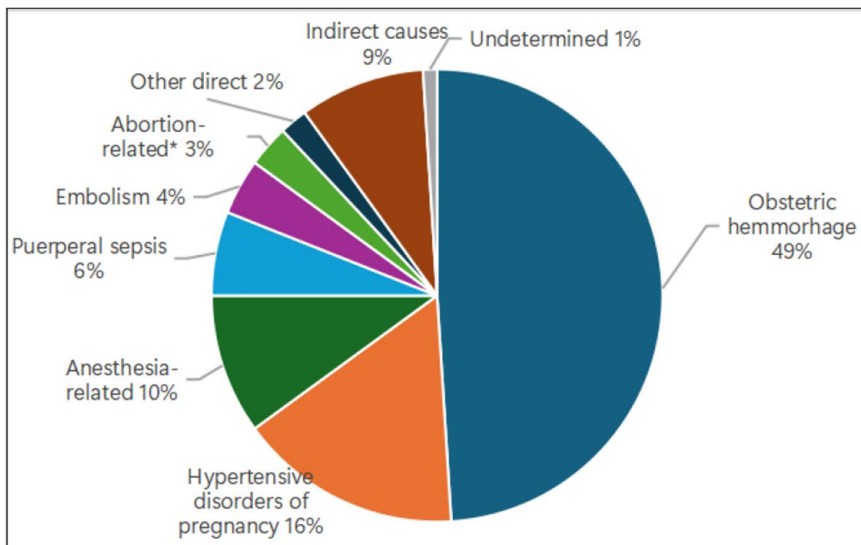

**Fig 1. Distribution of causes of deaths in 687 virtually reviewed maternal deaths in 2022–2023, Tanzania Mainland.**

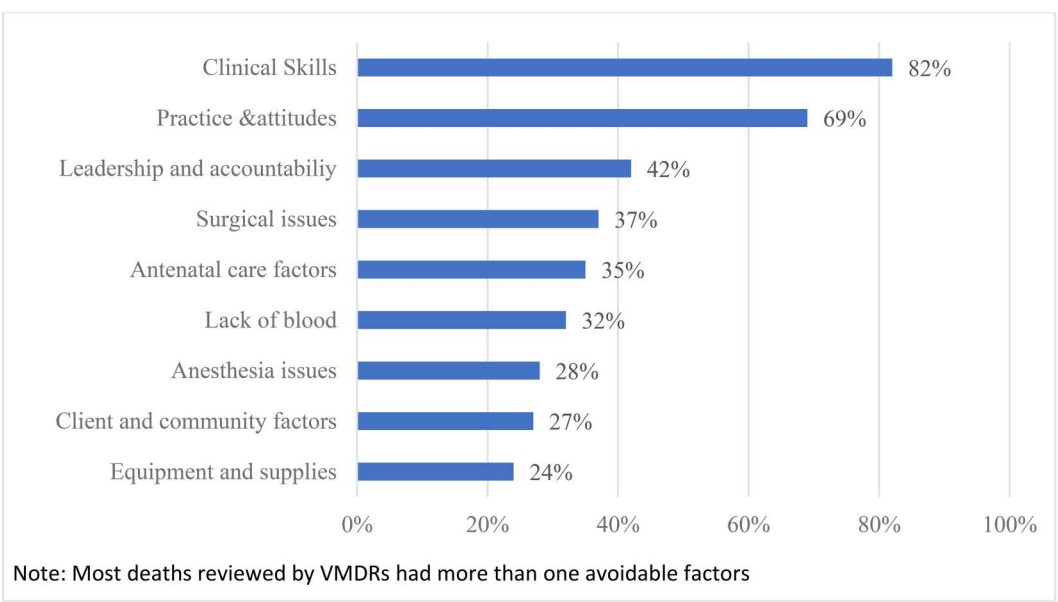

Note: Most deaths reviewed by VMDRs had more than one avoidable factors

**Fig 2. Main clinical and non-clinical modifiable factors in 687 virtually reviewed maternal deaths in 2022–2023, Tanzania Mainland.**

managed to reinforce adherence to standards of care in the management of intrauterine foetal deaths, use of misoprostol for induction of labour, removal of retained products of conception, and use of low molecular weight heparin to prevent and treat obstetric thromboembolism.

At the national level, the MoH, professional associations, and partners have intensified resource mobilisation and partnerships, creating a system for healthcare capacity building. This is shown by developing a national MNCH mentorship guide and supporting clinical attachments of healthcare providers from lower-level facilities to high-level facilities. The

Government has established mentorship in health facilities and commenced a virtual perinatal death review process and a weekly virtual Continuing Medical Education Forum (Table 2).

## Discussion

Studies and reports from more than a decade of MPDSR implementation in Tanzania have consistently shown limitations in the quality of reviews and weak accountability in follow-up actions, largely driven by insufficient reviewer capacity and inconsistent clinical oversight [16,18,20]. Against this background, this study set out to document the process and lessons learned from introducing VMDR as a strategy to strengthen the review system and address these persistent gaps.

Through the review of 687 maternal deaths, VMDR revealed recurrent gaps in clinical care, including limited clinical skills, suboptimal provider practices, and provider attitudes that affected the management of obstetric complications [28,29]. The process also demonstrated how VMDR supported greater accountability, as both managers and frontline healthcare providers were more responsive to the dysfunctions identified during the virtual reviews and more engaged in implementing corrective actions [30,31].

The introduction of VMDR strengthened the quality and consistency of maternal death reviews in Tanzania by widening access to specialist expertise, improving the identification of modifiable factors, and supporting more timely follow-up actions across levels of the health system. Similar benefits have been observed in other virtual and digital review initiatives. International virtual confidential reviews conducted across 11 low- and middle-income countries, including Ethiopia, Nigeria, Kenya, Ghana, South Africa, Zambia, Uganda, Pakistan, Bangladesh, Jordan, and Morocco, demonstrated that remote expert panels improved identification of clinical and health system gaps and strengthened learning and accountability in maternal death reviews [32]. In addition, telehealth-supported maternal health initiatives implemented in low-resource settings, including underserved areas of sub-Saharan Africa and South Asia, have shown how digital platforms can enhance access to specialist input and support timely clinical decision-making in maternal care [33]. Unlike these studies, which focused on discrete clinical domains or external review mechanisms, VMDR established a continuous, nationally coordinated platform embedded within the MPDSR system, enabling collaborative review, consistent interpretation of clinical events, and strengthened accountability.

Through this approach, VMDR created a structured environment where clinicians and managers at facility, district, regional and national levels could jointly analyse cases, clarify gaps in clinical care and system functioning, and agree on targeted actions. This aligns with recommendations that maternal death reviews should directly inform ongoing quality-improvement activities and not remain isolated discussion forums [11,24]. By linking case review with follow-up actions, VMDR helped translate MPDSR findings into concrete improvements, including reorganisation of staffing arrangements, strengthened referral pathways, targeted mentorship and supportive supervision, and mobilisation of specialist support.

Participation of senior clinical leaders and health system managers further reinforced implementation, as their presence strengthened accountability, heightened awareness of identified challenges and facilitated timely responses. These patterns mirror broader lessons from MPDSR implementation research across sub-Saharan Africa, where leadership engagement and structured review processes are consistently associated with improved responsiveness and more effective action [18,19]. Together, this evidence highlights the value of VMDR as a coordinated national mechanism that advances both learning and accountability, complementing global efforts to enhance the quality of maternal healthcare.

From the VMDR review process in Tanzania, maternal deaths predominantly occurred in health facilities and were largely associated with preventable or modifiable factors, including gaps in clinical knowledge, delayed recognition of complications, inconsistent adherence to standards of care, and inadequate supervision. These patterns closely mirror findings from MPDSR studies in Tanzania and other sub-Saharan African settings, where substandard clinical care, health system delays and poor decision making remain major contributors to maternal mortality [18,19,21,28,34]. In addition, this study identified provider attitude as an important contributor to suboptimal care, a finding that aligns with evidence from Kenya, Ghana and India showing that disrespect, poor communication and limited autonomy undermine women's

**Table 2. National Response using the VMDR Data.**

| S/N | Issue identified by VMDR | Response |
|---|---|---|
| **National level** | | |
| 1. | Mismatch in understanding between policymakers and implementers on overriding challenges in maternal health care | Policymakers have the same understanding as implementers that challenges on competence and leadership are key interventions for the health sector for now. |
| | | High-level National and Ministry leaders owned the results from RCA that over 80% of current maternal deaths are attributed to issues of competence, practice, attitude, and leadership. |
| 2. | Limited availability of consolidated, real-time data to support a strong, evidence-based investment case for maternal health financing Limited availability of consolidated, real-time data to support a strong, evidence-based investment case for maternal health financing | More financial resources have been mobilised and allocated to address the competence gap, hire human resources for health (HRH), and procure lifesaving commodities and equipment. |
| 3. | Limited quantities and quality of HRH in maternal healthcare | MOH started a mentorship plan where specialists from referral health go to primary health care facilities to work and mentor staff for at least two weeks. |
| | | Instituted a plan to have a panel of trainers for each region to support primary-level HFs. |
| | | More investment has been allocated to train service anaesthetists, anaesthesiologists, midwives, and obstetricians. |
| 4. | Inadequate readiness of the service delivery system to manage common obstetric complications due to a lack of providers' competency | Commencement of weekly virtual CME sessions to improve knowledge and skills in managing common obstetric complications. A total of 42 sessions were conducted from 20222023 |
| | | Supporting experts' teams to conduct visits to regions and facilities for follow-up on the implementation status of action plans agreed upon during virtual reviews. |
| | | Supporting experts' teams to visit regions and facilities with high burden/recurrences of specific causes of death for a detailed targeted clinical audit and RCA. |
| | | Development of the National Maternal, Newborn, and Child Health Mentorship guidelines and strategy to standardise capacity building for HCPs. |
| | | Design and commence implementation of structured clinical mentorship programs to build HCPs' capacity for managing common obstetric complications. |
| | | Sharing the findings and conducting high-level panel discussions with experts from medical training institutions on challenges of skills, practice, attitude, leadership, and stewardship of health service delivery |
| 5. | Persistence of PPH as the cause of maternal death | Commencement of review of national EmONC Job aids, learning resource Package to adopt the 2023 WHO Recommendations on Management of PPH, revised to accommodate findings of a groundbreaking multi-country and multicentric trial on early detection and management of PPH. |
| 6. | Full-strength use of technologies, medicines, and medical devices | Improved use of uterotonic guidelines in the induction and augmentation of labour, especially in preeclampsia/eclampsia, management of intrauterine foetal deaths, and postdate pregnancies. |
| | | Increase in use of heparin in post-caesarean section patients and other high-risk patients for venous thromboembolism features. |
| 7. | Persistent low quality of MDRs | Conduct training of the first Batch of 30 National MPDSR trainers across 10 priority regions to enable them to cascade these skills to peers in their home regions and facilities. |
| 8. | Lack of adherence to guidelines and SOPs | Commenced revision of the National EmONC learning resource package and Job aids to accommodate recent Global updates, e.g., E-motive Bundle in Management of PPH. |
| **Regional/Council level** | | |
| 1. | Lack of healthcare providers' skills in the management of obstetric complications, including the performance of obstetric surgeries | Conduct clinical attachments of healthcare providers with limitations in skills to tertiary facilities for a duration of four to 12 weeks. |
| | | Organise and support multi-disciplinary teams of experienced obstetricians/Gynaecologists to visit facilities for onsite mentorship on specific topics |
| | | Establishment and support of mobile emergency obstetric teams to promptly travel to respective facilities to attend emergencies |
| 2 | Limited number and quality of anaesthesia practitioners | Support tuition fees for clinical officers and nurse-midwives to undertake one year of anaesthesia training at tertiary facilities |
| | | Redistribution of anaesthetists from District Hospitals to Health centres lacking/in a critical shortage of anaesthetists. |

*(Continued)*

**Table 2.** (Continued)

| S/N | Issue identified by VMDR | Response |
|---|---|---|
| 3 | Lack of a functional ambulance | Repair of the ambulance using the district council financial resources |
| 4 | Limited quality of maternal death reviews | Commencement of virtual independent reviews at the regional level in some regions |
| 5 | Shortage of healthcare providers in newly established CEmONC Health centres | Re-distribution/reallocation of healthcare providers from district hospitals with better quantities of HRH |
| | | Recruitment of new healthcare providers on short-term contracts using local financial resources |
| 6 | Inadequate quality of EmONC services at newly established CEmONC sites | Form and support a team of experienced practitioners from Tertiary facilities to conduct targeted clinical audits on care systems, caesarean sections, and near-miss cases and respond accordingly to gaps identified |
| 7 | Inappropriate referral procedures, including a lack of skilled healthcare providers with equipment and supplies to support referrals | Ensure availability of an appropriately skilled cadre with needed resuscitation equipment to escort clients during referrals to ensure continuity of care |
| Facility level | | |
| 1. | Senior experts on Emergency roasters are staying at home at night in public hospitals | Re-arrangements made, including creating an enabling environment for teams on call to stay at the facility throughout the night and promptly respond to emergencies |
| 2 | Failure to adhere to standard operating procedures, guidelines, and SOPs | Printing, distribution, and orientation of healthcare providers on the use of the standard operating procedures and job aids |
| 3 | Lack of staff organization at the facilities, including work rosters and ward rounds | Reorganise staff through routine and emergency(on-call) duty rosters and commencement of routine daily ward rounds. |
| 4 | Frequent stockouts of routine and emergency medicines and supplies | Designate providers responsible for forecasting, routine stock management, and re-stocking (early ordering) of commodities |

Abbreviations: CEmONC, comprehensive emergency obstetric and neonatal care; CME, continuous medical education, EmONC, emergency obstetric and neonatal care; E-MOTIVE, early uterine massage, oxytocic drugs, tranexamic acid, intravenous fluids, and examination plus escalation; HCPs, health care professionals; HF, health facility; HRH, human resources for health; MoH, Ministry of Health; MPDSR, maternal and perinatal death surveillance and response; PPH, post-partum haemorrhage; RCA, root cause analysis; SOPs, standard operating procedures; ToT, training of trainers; WHO, World Health Organization.

experiences of facility-based maternity care and compromise quality [35]. Together, these findings highlight that improving maternal outcomes requires strengthening both technical competence and the behavioural dimensions of care, reinforcing the dual clinical and person-centred focus emphasised in the VMDR process.

The relatively high proportion of anaesthesia-related maternal deaths identified through VMDR may partly reflect stronger documentation for these cases, but it also suggests persistent gaps in anaesthesia skills and facility readiness for emergency obstetric care. Similar concerns have been raised in evidence from low and middle-income countries, where anaesthesia-related mortality, though often underreported, represents an important and preventable contributor to maternal deaths [36]. These findings highlight the need for targeted skills building, strengthened supervision and continued clinical mentorship for anaesthesia providers at all levels of the health system.

This study had some limitations. Case selection depended on the completeness of clinical documentation and the quality of prior facility or district reviews, which may have resulted in underrepresentation of cases with inadequate records, a challenge noted in other MPDSR assessments in Tanzania and comparable settings [21,37]. The reliance on facility-based deaths limited the assessment of community-level delays, including delays in seeking care or reaching a facility, which remain important contributors to maternal mortality in many sub-Saharan African contexts [27]. These limitations reflect long-standing structural challenges within MPDSR implementation rather than weaknesses of the VMDR model itself. They underscore the need for continued investment in accurate documentation, strengthened surveillance systems, and improved integration of community level information to ensure a more comprehensive understanding of preventable maternal deaths.

The distribution of VMDRs by cause is not representative. Still, it generally aligns with global, regional and national causes of MDS, where obstetric haemorrhage and pre-eclampsia are the leading causes [5,6,28]. One exception is the

high proportion of maternal deaths related to obstetric anaesthesia documented in VMDRs, which is several times higher than in the published literature. A recent systematic review found that anaesthesia accounted for an estimated 3.5% of direct maternal deaths in low- and middle-income countries [36]. Whereas the selection of cases submitted for virtual reviews may have influenced the percentage distribution of causes of maternal deaths, the true magnitude of anaesthesia-related deaths in Tanzania remains unclear.

Conducting VMDR consistently and regularly may help key players in maternal health care and related disciplines over-come their comfort zones and provide quality care through improved competence, practice, and attitudes. The success of VMDRs relies on regional, subnational, and national champions, including professional associations, who are dedicated to the entire review process and following up on recommendations.

The quality of documents used in reviews may directly impact the recommendations from maternal death audits. Missing or incomplete patient information and data on the care received may lead to unsatisfactory conclusions on the' causes and circumstances of maternal deaths and failure to identify substandard care and corrective actions correctly [36–38]. In VMDR, data abstraction is thorough, with clear event narration and documentation on the cause of death and underlying factors. If records are incomplete, an additional step is triggered via WhatsApp to reach out to all healthcare providers involved in the case management and complete missing information. This approach has enabled the generation of clear documentation on the cases that were reviewed and, in turn, enabled accurate descriptions of medical and non-medical causes that contributed to the death and improved the quality of recommendations. The same has influenced the government and stakeholders to adopt the best approaches to improve the quality of care, focusing on improving health-care providers' competency, practice and attitude through pre-service courses and in-service mentorship and supported supervision.

The selection of cases for VMDR is delegated to different regions. Therefore, the review team does not have control over the documentation and presentation of each case submitted for virtual review. Maternal deaths that are identified by regions have already been reviewed by health facilities and sometimes by district and regional committees. The selection of the VMDR cases may have been biased toward cases with complete medical information and documentation. Maternal deaths with incomplete or inadequate information or those whose care spanned over several wards or health facilities or required readmission may not be selected for VMDRs, which reduces the opportunity to learn from and formulate actions toward quality improvement targeting the most difficult cases. As VMDRs usually do not extend to maternal deaths outside health facilities, few community deaths were reviewed by the VMDR. Further, some facility MDR committees may not review maternal deaths due to indirect maternal causes, which may explain the low contribution of these causes among VMDR cases.

## Conclusion and recommendations

Maternal and Perinatal Death Surveillance and Response (MPDSR) remains a critical but complex undertaking in low-resource settings. Although widely implemented, many countries continue to face challenges in sustaining high-quality reviews and ensuring that findings translate into meaningful action. The WHO implementation guidance recognises that MPDSR must be adapted to the country context, with mechanisms that reflect the realities of each health system.

Tanzania's experience with Virtual Maternal Death Reviews (VMDR) demonstrates that structured digital platforms can reinforce MPDSR implementation by widening access to expert reviewers, improving the consistency of clinical analysis, and strengthening accountability for follow-up actions. The ability of VMDR to convene clinicians and managers from multiple levels of the health system in real time has created an environment where gaps in care can be clarified quickly, lessons shared more broadly, and recommendations acted upon with greater urgency. The sustained participation of national and regional leaders has also helped uphold core MPDSR principles of confidentiality and a non-punitive review culture, which have been essential for maintaining engagement.

Overall, VMDR has shown feasibility and acceptability as a routine review approach and has strengthened the link between case review and corrective action. Continued leadership commitment, investment in capacity building, and reinforcement of a supportive review culture will be key to preserving these gains. As countries explore innovative models to overcome persistent MPDSR implementation challenges, the Tanzanian VMDR experience offers a promising example of how digital solutions can enhance learning, improve clinical oversight, and promote accountability for maternal health outcomes. Further evaluation of VMDR in comparable settings would help build the evidence base for WHO and global partners to consider broader adoption of similar virtual review models across member states.

## Supporting information

**S1 File. VMDR Dataset.**
(XLSX)

**S2 File. Modifiable factors in VMDR.**
(DOCX)

## Acknowledgments

We thank our Ministers, Permanent Secretaries, Chief Medical Officer, Directors from MOH and PORALG, and professional bodies and associations for supporting VMDR. We sincerely thank the VMDR steering and coordination team for their tireless work to make this intervention run uninterrupted for three years from 2022 to date. The names these members are as follows; Sunday Alfred Dominico(MPDSR Focal, MOH), Felix Ambrose Bundala – Assistant Director (MPH), Saturin Manangwa – MOH (Nurse Midwife), Ziada Sellah – MOH Director Nurse Midwife Services, Upendo Mamchony – MOH (Nurse Midwife), Cecilia Msafiri – MOH (Nurse-Midwife specialist), Agness Mbio – MOH (Nurse Midwife), Faraja Jacob Mgeni – MOH (Nurse Midwife), Jackline Ndanshau – MOH (Nurse Midwife), Juma Daimon Nyakina – Bukoba Regional Referral Hospital (RRH) (OBGY), Paul John Sanka – Dodoma RRH (Nurse Midwife), Grace Aziza Machenje – MOH (Nurse Midwife), Naibu Samweli Mkongwa – MOH (MPH), Ulimbakisye Yoram Macdonald – MOH (OBGY), Victoria Lyimo – MOH (Nurse Midwife), Enock Mwantyala – MOH (MPH), Angela Leornard – MOH (Paediatricaian), Rachel Greyshom Yangwa – MOH (Nurse Midwife), Ephraim Kaphilimbi Shilla – MOH (Nurse Midwife), Fidea Obimbo – MOH (Nurse Midwife), Naomi Chamhene - - MOH (Nurse Midwife), Tito Amosi Chaula – Nkinga RH (OBSGY), Alfred Laison Mwakalebela – Iringa RRH (OBGY), Edwin Rwebugisa Lugazia – MUHAS (Anaesthesiologist), Donald Micah Maziku – Tosamaganga Hospital (OBSGY), George Rweyemamu Alcard – Mbeya RRH (OBSGY), Elias Asiwelo Kaminyoge – Mbeya (Meta) Zonal Hospital (OBGY), Chrisostom Clarence Lipingu – USAID-Afya Yangu/Jhpiego (OBGY), Rita Elias Lyamuya – Morogoro RRH (OBGY) Ali Said – MUHAS (OBGY), Elias Gregory Kweyamba – TTCIH (OBSGY), Berno Kasmir Mwambe – Marie Stopes Hospital (OBGY), France John Rwegoshora – Mbeya (Meta) Zonal Hospital – (OBGY), Titus Bonifas Mmasi – Monduli DC (OBGY), Shedrack Andrew Magambo – UDOM Hospital (OBGY), Patrick Joseph Kushoka – DCMC Hospital (OBSGY), John Deogratias Lawi – Iringa RRH (OBGY), Matilda Michael Ngarina – MNH (OBS/GYN), Pendo Samwel Mlay – KCMC (OBGY), Emmanuel Imani Ngadaya – Tosamaganga Hospital (OBGY), Chetan Pratapsinh Ramaiya – (OBGY), Nestory Andrew Mkenda – USAID-Afya Yangu (OBGY), Naomi Amon Mwamanenge – Aghakan Hospital (Neonatologist), Lucy Lawrence Mpayo – Neonatologist, Halima Mohamed Kassim – Dodoma RRH (Paediatrician), Joyce Stephen Gimonge – Geita RRH (Paediatrician), Emmanuel Rhubera Mafwiri – Njombe RRH (OBGY), Georgina Balyorugulu – Kamanga Hospital (Paediatrician), Yustina Adam Tizeba – Shinyanga RRH (Paediatrician), Tatu Seif Mbotoni – Iringa RRH (Paediatrician), Castory Jerome Mwanga – Simiyu RRH (Paediatrician), Beatrice Erastus Mwilike – MUHAS (Nurse midwife), Karoli Jerome Muna – Kondoa DH (OBGY), Enock Hashim Dolli – Bukoba RRH (Paediatrician), Mwanahawa Ramadhani Suba – Nkinga RH (Paediatrician), Amina Shomari Shabani – Morogoro RRH (NurseMidwife), Happyphania Mathew – Kibaha TC Hospital (Paediatrician), Jafari Bakari Lutavi – MNH (Nurse Midwife), Bahati Katembo

– Benjamin Mkapa Hospital (Nurse Midwife), Beatrice Bernard Kambuga – DIHAS Milembe (Nurse Midwife), Elizabeth Mkashabani – Paediatrician, Florentina Mashuda – CUHAS BMC (Paediatrician), Dismas Matovelo – CUHAS BMC (OBGY), Monica Mumwi – Ndanda RH (Paediatrician), Mary Sapali – Iringa RRH (Nurse Midwife), Nikolas Albert Sagumo Chotta – Dodoma RRH (Paediatrician), Lucy Mabada – TAMA (Nurse Midwife), Feddy Mwanga – TAMA (Nurse Midwife), Hawa Ngasongwa – Morogoro RRH (Paediatrician) Elisamia Ngowi – Aghakan Hospital (Paediatrician). Also, we would like to thank the Regional Council and Health Management Teams for their support in case identification, preparation, presentation, and follow-up of the implementation of VMDR action plans. We thank everyone who participated and provided input to VMDR from Tanzania and abroad. Without all of you, the VMDR would not be possible.

## Author contributions

**Conceptualization:** Ahmad Mohamed Makuwani, Florina Serbanescu.

**Data curation:** Ahmad Mohamed Makuwani, Sunday Alfred Dominico, Secilia Kapalata Ngweshemi, Golden Mwakibo Masika, Rose Mpembeni, Mzee Masumbuko Nassoro, Phineas Sospeter, Matilda Ngarina, Edwin Lugazia, Pius Muzzazzi, Beatrice Mwilike, Florina Serbanescu.

**Formal analysis:** Ahmad Mohamed Makuwani, Sunday Alfred Dominico, Secilia Kapalata Ngweshemi, Mzee Masumbuko Nassoro, Phineas Sospeter, Habib Ismail, Matilda Ngarina.

**Funding acquisition:** Ahmad Mohamed Makuwani.

**Investigation:** Ahmad Mohamed Makuwani, Tumaini Nagu, Matilda Ngarina, Edwin Lugazia, Pius Muzzazzi, Beatrice Mwilike.

**Methodology:** Ahmad Mohamed Makuwani, Sunday Alfred Dominico, Secilia Kapalata Ngweshemi, Golden Mwakibo Masika, Rose Mpembeni, Charles Ameh, Mzee Masumbuko Nassoro, Habib Ismail, Edwin Lugazia, Florina Serbanescu.

**Project administration:** Ahmad Mohamed Makuwani, Secilia Kapalata Ngweshemi, Golden Mwakibo Masika, Tumaini Nagu.

**Resources:** Ahmad Mohamed Makuwani.

**Software:** Sunday Alfred Dominico, Rose Mpembeni, Mzee Masumbuko Nassoro, Phineas Sospeter, Florina Serbanescu.

**Supervision:** Secilia Kapalata Ngweshemi, Rose Mpembeni, Mzee Masumbuko Nassoro, Habib Ismail, Tumaini Nagu, Beatrice Mwilike.

**Validation:** Ahmad Mohamed Makuwani, Sunday Alfred Dominico, Golden Mwakibo Masika, Rose Mpembeni, Charles Ameh, Mzee Masumbuko Nassoro, Habib Ismail, Matilda Ngarina, Edwin Lugazia, Florina Serbanescu.

**Visualization:** Ahmad Mohamed Makuwani, Rose Mpembeni, Mzee Masumbuko Nassoro, Habib Ismail, Florina Serbanescu.

**Writing – original draft:** Ahmad Mohamed Makuwani.

**Writing – review & editing:** Ahmad Mohamed Makuwani, Sunday Alfred Dominico, Golden Mwakibo Masika, Rose Mpembeni, Charles Ameh, Phineas Sospeter, Habib Ismail, Matilda Ngarina, Edwin Lugazia, Pius Muzzazzi, Beatrice Mwilike, Florina Serbanescu.

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
