## [Decision Letter · Decision Letter 0]

3 Apr 2025

PONE-D-25-09586
Using national virtual maternal death reviews to improve the quality of care during Pregnancy, labour and birth, and postpartum in Tanzania
PLOS ONE

Dear Dr. Makuwani,

Thank you for submitting your manuscript to PLOS ONE. After careful consideration, we feel that it has merit but does not fully meet PLOS ONE’s publication criteria as it currently stands. Therefore, we invite you to submit a revised version of the manuscript that addresses the points raised during the review process.

We look forward to receiving your revised manuscript.

Kind regards,

Joyce Jebet Cheptum

Academic Editor

PLOS ONE

Journal Requirements:

“One author from US CDC supported data analysis and set up of the manuscript”

4. One of the noted authors is a group or consortium “Tanzania Virtual MPDSR Steering committee”.

In addition to naming the author group, please list the individual authors and affiliations within this group in the acknowledgments section of your manuscript. Please also indicate clearly a lead author for this group along with a contact email address.

6. Please amend your manuscript to include your abstract after the title page.

**Additional Editor Comments:**

Beef up the methodology section, especially the study design. It is not adequate to just mention quantitative and qualitative.

Reviewers' comments:

Reviewer's Responses to Questions

**Comments to the Author**

1. Is the manuscript technically sound, and do the data support the conclusions?

Reviewer #1: Yes

Reviewer #2: Yes

Reviewer #3: Partly

2. Has the statistical analysis been performed appropriately and rigorously?

Reviewer #1: Yes

Reviewer #2: Yes

Reviewer #3: Yes

3. Have the authors made all data underlying the findings in their manuscript fully available?

Reviewer #1: No

Reviewer #2: Yes

Reviewer #3: Yes

4. Is the manuscript presented in an intelligible fashion and written in standard English?

Reviewer #1: Yes

Reviewer #2: Yes

Reviewer #3: Yes

5. Review Comments to the Author

Reviewer #1: The manuscript is well-written with very few typographical errors (eg data "achieved" instead of "archived", "is" instead of "was"). These were highlighted in the text as "comments." The work addresses a very important intervention in obstetrics practice in line with WHO recommendations which is commendable.

Reviewer #2: Thank you for asking me to review this research which is an excellent innovation and appropriate for MPDSR

I wish to note the followings:

1. The title is apt.

2. Discussion: The first two sentences are unnecessary repetition of results/findings already highlighted, hence should be rephrased.

3. Conclusion: MPDSR should be written in full. Acronyms even when previously defined are best avoided in starting a sentence.

4. A few words were in British and some in American English. It is best to use one consistently in a manuscript.

5. I have added a few annotations to corrections

These are minor corrections and the article can be accepted for publication afterwards.

Thank you

Reviewer #3: Dear Author,

Your manuscript holds significant value. Below are some suggestions to help enhance your research:

1. In the abstract, clearly articulate the necessity of the research.

2. In the introduction, outline the innovative aspects of your research in the final paragraph.

3. At the conclusion of the discussion, incorporate suggestions for future studies.

6. PLOS authors have the option to publish the peer review history of their article (what does this mean?). If published, this will include your full peer review and any attached files.

Reviewer #1: No

Reviewer #2: **Yes:**Matthias Gabriel Abah

Reviewer #3: No

---

## [Author Response · Author response to Decision Letter 1]

29 May 2025

See the attachments; revised manuscript without comments, revised manuscript with changes, and letter with responses on the revision.

---

## [Decision Letter · Decision Letter 1]

12 Aug 2025

PONE-D-25-09586R1
Using national virtual maternal death reviews to improve the quality of care during Pregnancy, labour and birth, and Postpartum in Tanzania
PLOS ONE

Dear Dr. Makuwani,

Thank you for submitting your manuscript to PLOS ONE. After careful consideration, we feel that it has merit but does not fully meet PLOS ONE’s publication criteria as it currently stands. Therefore, we invite you to submit a revised version of the manuscript that addresses the points raised during the review process.

We look forward to receiving your revised manuscript.

Kind regards,

Joyce Jebet Cheptum

Academic Editor

PLOS ONE

Journal Requirements:

Reviewers' comments:

Reviewer's Responses to Questions

**Comments to the Author**

1. If the authors have adequately addressed your comments raised in a previous round of review and you feel that this manuscript is now acceptable for publication, you may indicate that here to bypass the “Comments to the Author” section, enter your conflict of interest statement in the “Confidential to Editor” section, and submit your "Accept" recommendation.

Reviewer #4: All comments have been addressed

Reviewer #5: All comments have been addressed

Reviewer #6: All comments have been addressed

Reviewer #7: (No Response)

2. Is the manuscript technically sound, and do the data support the conclusions?

Reviewer #4: Yes

Reviewer #5: Yes

Reviewer #6: Yes

Reviewer #7: Partly

3. Has the statistical analysis been performed appropriately and rigorously?

Reviewer #4: Yes

Reviewer #5: Yes

Reviewer #6: Yes

Reviewer #7: No

4. Have the authors made all data underlying the findings in their manuscript fully available?

Reviewer #4: Yes

Reviewer #5: No

Reviewer #6: Yes

Reviewer #7: Yes

5. Is the manuscript presented in an intelligible fashion and written in standard English?

Reviewer #4: Yes

Reviewer #5: Yes

Reviewer #6: Yes

Reviewer #7: Yes

6. Review Comments to the Author

Reviewer #4: The authors have tried in making most of the corrections and highlighting the corrections and showing a table of the reviewers comments with the corrections made.

Section on Data ethics management

"These data were achieved and hence did not require individual clearance." Is the word achieved or archived please check?

Reviewer #5: well written. PPH has consistently been major cause of maternal mortality globally. The online surveillance should however be assessed for feasibility and cost-effectiveness

Reviewer #6: Thanks to the authors to adress such a vital issue as an estimate of the progress of health services both in developing as well as developed counteries, the outcomes of these studies will be a guide to improve health sevices within the country.

Reviewer #7: (No Response)

7. PLOS authors have the option to publish the peer review history of their article (what does this mean?). If published, this will include your full peer review and any attached files.

Reviewer #4: **Yes:**Maryam Jamila Ali

Reviewer #5: **Yes:**Emmanuel Ajuluchukwu Ugwa PhD

Reviewer #6: **Yes:**Mohsen M A Abdelhafez

Reviewer #7: **Yes:**Mercy Monde Imakando

---

## [Author Response · Author response to Decision Letter 2]

27 Aug 2025

Comments addressed as per attached file: reviewers comment responses

---

## [Decision Letter · Decision Letter 2]

4 Nov 2025

PONE-D-25-09586R2
Using national virtual maternal death reviews to improve the quality of care during Pregnancy, labour and birth, and Postpartum in Tanzania
PLOS ONE

Dear Dr. Makuwani,

Thank you for submitting your manuscript to PLOS ONE. After careful consideration, we feel that it has merit but does not fully meet PLOS ONE’s publication criteria as it currently stands. Therefore, we invite you to submit a revised version of the manuscript that addresses the points raised during the review process.

We look forward to receiving your revised manuscript.

Kind regards,

Joyce Jebet Cheptum

Academic Editor

PLOS ONE

Journal Requirements:

Reviewers' comments:

Reviewer's Responses to Questions

**Comments to the Author**

1. If the authors have adequately addressed your comments raised in a previous round of review and you feel that this manuscript is now acceptable for publication, you may indicate that here to bypass the “Comments to the Author” section, enter your conflict of interest statement in the “Confidential to Editor” section, and submit your "Accept" recommendation.

Reviewer #3: (No Response)

Reviewer #7: All comments have been addressed

2. Is the manuscript technically sound, and do the data support the conclusions?

Reviewer #3: (No Response)

Reviewer #7: Yes

3. Has the statistical analysis been performed appropriately and rigorously?

Reviewer #3: (No Response)

Reviewer #7: Yes

4. Have the authors made all data underlying the findings in their manuscript fully available?

Reviewer #3: (No Response)

Reviewer #7: Yes

5. Is the manuscript presented in an intelligible fashion and written in standard English?

Reviewer #3: (No Response)

Reviewer #7: Yes

6. Review Comments to the Author

Reviewer #3: Dear Author,

The following comments are intended to help improve your research:

1. The necessity of the study is not clearly articulated in the abstract.

2. In the methods section of the abstract, please specify the criteria for the study.

3. In the last paragraph of the introduction, highlight the innovations present in your research.

4. In the methods section, provide a more detailed explanation of the data collection and sampling methods.

5. Strengthen the discussion section.

6. Many of the references are outdated; please use sources from the last five years.

Reviewer #7: (No Response)

7. PLOS authors have the option to publish the peer review history of their article (what does this mean?). If published, this will include your full peer review and any attached files.

Reviewer #3: No

Reviewer #7: **Yes:**Mercy Monde Imakando

---

## [Author Response · Author response to Decision Letter 3]

25 Nov 2025

All the six comments have been addressed as shown in the rebuttal letter

---

## [Decision Letter · Decision Letter 3]

4 Jan 2026

PONE-D-25-09586R3
Using national virtual maternal death reviews to improve the quality of care during Pregnancy, labour and birth, and Postpartum in Tanzania
PLOS One

Dear Dr. Makuwani,

Thank you for submitting your manuscript to PLOS ONE. After careful consideration, we feel that it has merit but does not fully meet PLOS ONE’s publication criteria as it currently stands. Therefore, we invite you to submit a revised version of the manuscript that addresses the points raised during the review process.

We look forward to receiving your revised manuscript.

Kind regards,

Joyce Jebet Cheptum

Academic Editor

PLOS One

Journal Requirements:

Additional Editor Comments (if provided):

Dear Authors,

Please address the few comments related to maternal mortality in Tanzania

Reviewers' comments:

Reviewer's Responses to Questions

**Comments to the Author**

1. If the authors have adequately addressed your comments raised in a previous round of review and you feel that this manuscript is now acceptable for publication, you may indicate that here to bypass the “Comments to the Author” section, enter your conflict of interest statement in the “Confidential to Editor” section, and submit your "Accept" recommendation.

Reviewer #8: All comments have been addressed

Reviewer #9: (No Response)

2. Is the manuscript technically sound, and do the data support the conclusions?

Reviewer #8: Yes

Reviewer #9: Yes

3. Has the statistical analysis been performed appropriately and rigorously?

Reviewer #8: Yes

Reviewer #9: Yes

4. Have the authors made all data underlying the findings in their manuscript fully available?

Reviewer #8: Yes

Reviewer #9: Yes

5. Is the manuscript presented in an intelligible fashion and written in standard English?

Reviewer #8: Yes

Reviewer #9: Yes

6. Review Comments to the Author

Reviewer #8: The author did all changes,

the manuscript looks better now

accepted for publication

all the best

Reviewer #9: 1. What is the maternal mortality ratio for Tanzania? It did not reflect anywhere in the document.

2. Page 8. Since the three-delay model was put into consideration, how are modifiable community and family-oriented gaps that contributed to the mortality put into consideration?

3. Page 9. “Data not shown” How does this relate to places where the mortality occurred? What is the weighed percentage?

4. Page 9The statement “Due to rotating schedule of region…. The rotation does not adequately explain why region with fewer death had more reviews. Is it that regions with fewer death were given more opportunity? Did the rotation favored them more?

5. Page 12 – “Lack of a strong data backed investment case to advocate for financial resources in maternal healthcare “. The response documented regarding this gap seems inappropriate. Before funds can be duly allocated, there is need for adequate information or data for human resources. Allocating funds without strong data could result in inappropriate allocation,

6. Page 16.. “Similar benefits have been observed in other virtual and digital review initiatives”. Can the countries be cited and referenced?

7. Page 17 “ Most maternal deaths reviewed through VMDR occurred in health facilities and were linked to preventable or modifiable factors, including gaps in clinical knowledge, delayed recognition of complications, inconsistent adherence to standards of care, and inadequate supervision”. Can the context be included? For example, “From review, most maternal death…………”. It will help establish the statement as a context for comparison with Tanzania.

7. PLOS authors have the option to publish the peer review history of their article (what does this mean?). If published, this will include your full peer review and any attached files.

Reviewer #8: **Yes:**Zalikha Al-Marzouqi

Reviewer #9: **Yes:**Oluwadamilola Olaogun

---

## [Author Response · Author response to Decision Letter 4]

6 Jan 2026

5th January 2026

Dear Joyce Jebet Cheptum

Academic Editor

PLOS ONE

Dear Editor,

On behalf of all co-authors, I would like to thank you and reviewers for a big interest shown. careful assessment of our manuscript and for the constructive feedback provided throughout the review process, including the most recent comments received on 5 January 2026. We are encouraged by the positive evaluations confirming that the manuscript is technically sound, clearly written, and supported by appropriate data and analyses. We particularly appreciate the acknowledgement that the revised manuscript has improved substantially and that all previously raised comments have been adequately addressed.

In this revision, we have carefully considered and responded to all additional comments raised, including clarifications on the national maternal mortality context, application of the three-delay model, data presentation, interpretation of regional review patterns, framing of health financing gaps, citation of comparable international experiences, and contextualisation of key findings.

Detailed point-by-point responses to each comment are provided in the accompanying response table, with a clear indication of the corresponding revisions and their locations in the manuscript. We are grateful for the opportunity to revise and strengthen the manuscript and appreciate the Editor’s and Reviewers’ guidance throughout this process.

6.Review Comments to the Author

Reviewer Comment Response

Editor Dear Authors,

please address the few comments related to maternal mortality in Tanzania We have addressed this and other comments

Reviewer #8:  The author did all changes, the manuscript looks better now. Accepted for publication.All the best

Thank you very much for the positive and encouraging feedback.

Reviewer #9:  1. What is the maternal mortality ratio for Tanzania? It did not reflect anywhere in the document. Thank you for this important observation. Tanzania’s maternal mortality ratio is 104 maternal deaths per 100,000 live births, based on the 2022 Tanzania Demographic and Health Survey and Malaria Indicator Survey (TDHS - MIS). We acknowledge that this figure was not explicitly stated in the original version of the manuscript.

We have now revised the introduction to include the national maternal mortality ratio and appropriate citation, to better contextualize the burden of maternal mortality in Tanzania and strengthen the background of the study.

2. Page 8. Since the three-delay model was put into consideration, how are modifiable community and family-oriented gaps that contributed to the mortality put into consideration? Thank you for this comment. We have revised the manuscript and Figure 2 to explicitly capture client and community-related contributing factors within the application of the three-delay model. These factors accounted for 27% of the reviewed maternal deaths and include delayed care-seeking, household decision-making constraints, inadequate birth preparedness, and community - level referral challenges, as presented in the results section.

3. Page 9. “Data not shown” How does this relate to places where the mortality occurred? What is the weighed percentage? Thank you for pointing this. The phrase “data not shown” was included in error. All data on the place where maternal deaths occurred are fully presented in the manuscript. We have revised the text to remove this phrase and clarify the presentation accordingly

4. Page 9 - The statement “Due to rotating schedule of region…. The rotation does not adequately explain why region with fewer death had more reviews. Is it that regions with fewer death were given more opportunity? Did the rotation favoured them more? Thank you for this observation. We have revised the text to clarify that the higher proportion of deaths reviewed in regions with fewer maternal deaths reflects the structured rotation system, in which each region is allocated scheduled opportunities to present cases irrespective of the absolute number of deaths. As a result, regions with fewer deaths were able to review a larger proportion of their notified cases, rather than being favoured by the process. The revised text now makes this explicit to avoid misinterpretation.

5. Page 12 – “Lack of a strong data backed investment case to advocate for financial resources in maternal healthcare “. The response documented regarding this gap seems inappropriate. Before funds can be duly allocated, there is need for adequate information or data for human resources. Allocating funds without strong data could result in inappropriate allocation,

Thank you for this important clarification. We agree that allocation of financial resources should be guided by robust data and not precede it. We have revised the text to clarify that VMDR findings strengthened the data-backed investment case for maternal health by generating actionable evidence on gaps in human resources, competencies, and system readiness, which then informed prioritisation and allocation of resources. The revised text now emphasises evidence-informed decision-making rather than allocation in the absence of data.

6. Page 16.. “Similar benefits have been observed in other virtual and digital review initiatives”. Can the countries be cited and referenced? Thank you for this comment. We have revised the manuscript to explicitly cite and reference the countries where similar virtual and digital review initiatives have been implemented. The revised text now specifies countries involved in international virtual confidential reviews and contextualises telehealth - supported maternal health initiatives in low-resource settings, with appropriate references added in the discussion section.

7. Page 17 “ Most maternal deaths reviewed through VMDR occurred in health facilities and were linked to preventable or modifiable factors, including gaps in clinical knowledge, delayed recognition of complications, inconsistent adherence to standards of care, and inadequate supervision”. Can the context be included? For example, “From review, most maternal death…………”. It will help establish the statement as a context for comparison with Tanzania. Thank you for this helpful suggestion. We have revised the sentence to explicitly situate the statement within the VMDR review context in Tanzania, clarifying that the observation is based on findings from the reviewed cases. The revised wording now begins with “From the VMDR review process in Tanzania…”, thereby establishing a clear contextual basis for comparison with findings from other settings.

Sincerely

Ahmad M. Makuwani

Corresponding author

---

## [Decision Letter · Decision Letter 4]

27 Feb 2026

Using national virtual maternal death reviews to improve the quality of care during Pregnancy, labour and birth, and Postpartum in Tanzania

PONE-D-25-09586R4

Dear Dr. Makuwani,

We’re pleased to inform you that your manuscript has been judged scientifically suitable for publication and will be formally accepted for publication once it meets all outstanding technical requirements.

Kind regards,

Joyce Jebet Cheptum

Academic Editor

PLOS One

Additional Editor Comments (optional):

Reviewers' comments:

Reviewer's Responses to Questions

**Comments to the Author**

1. If the authors have adequately addressed your comments raised in a previous round of review and you feel that this manuscript is now acceptable for publication, you may indicate that here to bypass the “Comments to the Author” section, enter your conflict of interest statement in the “Confidential to Editor” section, and submit your "Accept" recommendation.

Reviewer #9: All comments have been addressed

2. Is the manuscript technically sound, and do the data support the conclusions?

Reviewer #9: Yes

3. Has the statistical analysis been performed appropriately and rigorously?

Reviewer #9: Yes

4. Have the authors made all data underlying the findings in their manuscript fully available?

Reviewer #9: Yes

5. Is the manuscript presented in an intelligible fashion and written in standard English?

Reviewer #9: Yes

6. Review Comments to the Author

Reviewer #9: (No Response)

7. PLOS authors have the option to publish the peer review history of their article (what does this mean?). If published, this will include your full peer review and any attached files.

Reviewer #9: No

---

## [Editor Report · Acceptance letter]

PONE-D-25-09586R4

PLOS One

Dear Dr. Makuwani,

I'm pleased to inform you that your manuscript has been deemed suitable for publication in PLOS One. Congratulations! Your manuscript is now being handed over to our production team.

Kind regards,

on behalf of

Dr. Joyce Jebet Cheptum

Academic Editor

PLOS One